# Taxonomy, Morphological and Molecular Identification of the Potato Cyst Nematodes, *Globodera pallida* and *G. rostochiensis*

**DOI:** 10.3390/plants10010184

**Published:** 2021-01-19

**Authors:** John Wainer, Quang Dinh

**Affiliations:** Agriculture Victoria Research, Department of Jobs, Precincts and Regions, Bundoora, VIC 3086, Australia; quang.dinh@agriculture.vic.gov.au

**Keywords:** PCN, potato cyst nematode, *Globodera*, taxonomy, detection, morphology, molecular identification, PCR

## Abstract

The scope of this paper is limited to the taxonomy, detection, and reliable morphological and molecular identification of the potato cyst nematodes (PCN) *Globodera pallida* and *G. rostochiensis*. It describes the nomenclature, hosts, life cycle, pathotypes, and symptoms of the two species. It also provides detailed instructions for soil sampling and extraction of cysts from soil. The primary focus of the paper is the presentation of accurate and effective methods to identify the two principal PCN species.

## 1. Introduction

Potato cyst nematodes (PCN) are damaging soilborne quarantine pests of potato and other solanaceous crops worldwide [1,2]. The two most damaging species, *G. pallida* (Stone, 1973) Behrens, 1975, the pale or white cyst nematode, and *G. rostochiensis* (Wollenweber, 1923) Behrens, 1975, the golden cyst nematode, have proved to be highly adaptive at exploiting new environments, being passively transported, undetected across borders, in intimate association with tubers of their major host, the potato. *Globodera* species feeding on potato also include *G. ellingtonae*, restricted to Chile, Argentina, and two states in northwest USA [3,4,5] and *G. leptonepia* (Cobb and Taylor, 1953) Skarbilovich, 1959 found only once in a ship-borne consignment of potatoes [6,7].

Potato cyst nematodes are obligate sedentary endoparasites that can cause stunting of plants, reduce yields, and sometimes lead to complete crop failure. PCN causes losses of 9% of total potato production in Europe and can cause total losses in other parts of the world when no control strategies are employed [8]. When PCN populations are high in the field, potato yields can be less than the tonnage per unit area of the planted seed [9,10]. PCN presents formidable problems to farmers, advisors, and policy makers due to their small size and cryptic nature within large volumes of soil, their extreme specialization and intimate association with their host, and their adaptation for long-term survival in the soil in the absence of a suitable host. In fact, PCN is recognized throughout the temperate regions of the world as one of the most difficult crop pests to control [11].

As internationally recognized plant-quarantine organisms, efficient sampling and detection methods of PCN are critical to the effective management of these pests in both emergency response and on-going control situations [12,13,14,15]. Cysts are the dead remnants of female nematodes and contain hundreds of eggs; they can survive in soil without a host for 20 years or more [16]. *Globodera rostochiensis* and *G. pallida* are closely related species and difficult to be distinguished from each other solely based on morphology. The European and Mediterranean Plant Protection Organization has published a diagnostic protocol for the two species [17].

## 2. Nomenclature

Phylum: Nematoda Diesing, 1861, Order: Rhabditida Chitwood, 1933, Suborder: Tylenchina Chitwood, 1950, Family: Heteroderidae Filip’ev & Schuurmans Stekhoven, 1941, Genus: *Globodera* (Skarbilovich, 1959) Behrens, 1975.

*Globodera pallida* synonyms:

*Heterodera rostochiensis* Wollenweber, 1923 in partim

*Heterodera pallida* Stone, 1973

*Heterodera (Globodera) pallida* Stone, 1973

*Globodera pallida* (Stone, 1973) Mulvey and Stone, 1976.

*Globodera pallida* common names:

English: PCN, white potato cyst nematode, pale potato cyst nematode; French: nématode blanc de la pomme de terre; Spanish: nemátodo quiste blanco de la papa.

*Globodera rostochiensis* synonyms:

*Heterodera rostochiensis* Wollenweber, 1923

*Heterodera schachtii rostochiensis* Wollenweber, 1923

*Heterodera schachtii solani* Zimmermann, 1927

*Heterodera solani* Zimmermann, 1927

*Heterodera* (*Globodera*) *rostochiensis* (Wollenweber, 1923) Skarbilovich, 1959

*Heterodera pseudorostochiensis* Kirjanova, 1963

*Globodera pseudorostochiensis* (Kirjanova, 1963) Mulvey and Stone, 1976

*Globodera rostochiensis* (Wollenweber, 1923) Mulvey and Stone, 1976

*Globodera arenaria* Chizhov, Udalova and Nasonova, 2008.

*Globodera rostochiensis* common names:

English: PCN, yellow potato cyst nematode, golden potato cyst nematode, golden nematode, potato root eelworm; French: anguillule a kyste de la pomme de terre, anguillule des racines de la pomme de terre, nématode doré, nématode doré de la pomme de terre; German: kartoffelnematode; Spanish: nemátodo dorado.

## 3. Hosts

PCN hosts are restricted to the nightshade family Solanaceae. The most important host is *Solanum tuberosum* (potato) although other agronomic crops such as *Solanum lycopersicum* (tomato) and *Solanum melongena* (eggplant) are also attacked [18]. Up to 90 *Solanum* spp. And their hybrids can be PCN hosts including some weed species. These include *Datura* spp. (devil’s trumpets), *Hyoscyamus niger* (henbane), *Nicotiana acuminata* (manyflower tobacco), *Physalis* spp. (husk tomatoes), *Physochlaina orientalis* (oriental physochlaina), *Salpiglossis* sp. (painted tongue), *Capsicum annuum* (chili pepper), and *Jaltomata procumbens* (creeping false holly) [19,20]. For a more complete list of PCN hosts, see [21,22,23,24].

Vermiform juveniles and adult PCN cysts can be found either in the soil or attached to roots or tubers, whereas adult males are found exclusively in the soil.

## 4. Life Cycle

The PCN cyst is the hardened dead body of a female and protects the eggs within. It is spheroid with a short neck. The female *G. rostochiensis* changes during maturation from white to yellow and then into brown cysts, whereas *G. pallida* changes from creamy white directly to brown. Cysts are highly resistant and long-lived and can be readily spread, mostly in association with soil, by human activities [25].

After infective juvenile nematodes hatch, they can disperse in the soil a distance of about 1 m and infect plants by entering a root near the growing tip. The nematode becomes sedentary, establishing a feeding site by modifying plant cells which then provide nutrients. Infested potato plants have a reduced root system and poor productivity [26]. Plant death can occur [27].

The lifecycle of PCN (Figure 1) can be described as follows:

A period of 38–48 days (depending on soil temperature) is required for PCN to complete its life cycle [28]. Nematodes reproduce sexually; males are attracted to females by a pheromone sex attractant. Nematodes may mate several times. After mating, each female produces approximately 200–500 eggs [29], dies, and the cuticle of the dead female forms a cyst. Eggs mostly remain dormant within the cyst until receiving a hatching stimulus (i.e., specific chemical released by host plant roots). PCN eggs can remain dormant and viable within the cyst for at least 30 years [16] and are resistant to nematicides [11].

When soil temperatures are warm enough (above 10 °C) [30], and hatching stimuli are received [31], second-stage juveniles hatch from the eggs, escape from the cyst, and migrate towards the host plant roots. Egg hatching is stimulated by host root diffusate, but not all eggs hatch (60–80%); by comparison only about 5% will hatch in water. Some eggs do not hatch until subsequent years [2].

Juveniles penetrate roots where they begin to feed. Host plant cells within the root cortex are stimulated to form specialized cells (syncytia) which transfer nutrients to the nematodes. After feeding commences, the juvenile grows and undergoes three more moults to become an adult. Females grow and become round, breaking through the roots and exposing the posterior portion of their body to the external environment.

Male juveniles remain active, feeding on the host plant until maturity, at which time they stop feeding, become vermiform, and seek females [32]. Adult males do not feed. Sex is determined by food supply—more juveniles develop into males under adverse conditions and heavy infestations.

## 5. Pathotypes

Pathotypes (or virulence groups) of PCN are characterized by ability to multiply on certain clones and hybrids of *Solanum* spp. Both species of PCN have several pathotypes under several different schemes [1,33,34,35,36]. Under the European scheme [35], there are five pathotypes (Ro1–Ro5) for *G. rostochiensis* and three (Pa1–Pa3) for *G. pallida*. A wide range of commercial potato cultivars currently available carry the H1 gene that confers near complete resistance to the Ro1 and Ro4 pathotypes [37] and other genes (e.g., Gro1) confer resistance to all *G. rostochiensis* pathotypes. Although various genes confer a degree of resistance to *G. pallida*, complete resistance is not known, which means that some multiplication of the nematode is possible for most commercial cultivars [38]. The term “pathotype” is now considered too general, as many PCN populations cannot be assigned conclusively to pathotypes [17]. Any population showing signs of a new virulence should be tested as soon as possible.

## 6. Symptoms

Symptoms of PCN infestation are not specific and may not be apparent even when crop yield is significantly reduced. At high densities, patches of poor growth can occur in potato crops, sometimes with yellowing, wilting, or necrosis of the foliage. These symptoms may be caused by many other plant pathogens, including other nematodes, and should not be considered proof of PCN presence. If there are clear patches of stunting, plants should be lifted for a visual check for cysts on the roots. This is only possible for a short time at the appropriate stage of the crop; as young females mature into cysts they are easily detached when lifting plants.

When infested plants are lifted carefully, the swollen females or the cysts appear as small bead-like objects attached to the roots and can be easily seen with the naked eye. With severe infestations, cysts may be seen on the surface of tubers or stolons.

A cyst that changes during maturation from white to yellow and then into brown is *G. rostochiensis* while one which changes from creamy white directly to brown is *G. pallida*. Note that this feature can only be used at the appropriate stage of the life cycle: young cysts of both species are white or cream, and mature cysts of both species are brown.

## 7. Soil Sampling

Visual symptoms alone cannot be used to identify the presence of PCN in a potato crop. There are two methods available to sample fields for PCN: (1) taking soil samples and processing them in the laboratory; or (2) lifting plants and examining their roots for females or cysts in either the field or laboratory. The latter method has been used to detect low populations, which may have been undetectable by soil sampling [39]. However, plant sampling is extremely labor intensive, and plants are available only during a part of the year or cropping cycle, whereas viable cysts can remain in the soil for many years. Soil samples must be large enough to achieve the required accuracy and sensitivity and must also be derived from many points in the field to ensure that they are representative of the area. Been and Schomaker [40] emphasized the importance of sample point spacing to the probability of detection of PCN cysts in a field. To achieve a 90% average probability of detection, grid sampling at 5 m spacing with 52 g cores (total sample size 6.9 kg/0.33 ha) was recommended as being the best compromise for minimizing sample size and maximizing detection probability, while minimizing time needed to collect and process the samples. This recommendation is based on detecting the minimum abundance of cysts which will cause crop losses, rather than the presence or absence of PCN. The level of sampling depends on the aim: delimiting surveys for biosecurity reasons require a relatively high level of accuracy (i.e., a high probability of detection) whereas routine sampling, for example of seed potato crops, is generally done at a lower level of accuracy and probability of detection.

## 8. Cyst Extraction

*Globodera* spp. Juveniles and adult males can be extracted from soil by general nematode extraction methods such as Whitehead Trays or the more efficient differential flotation [41,42]. An additional cyst extraction on the soil is desirable because a combination of cyst and juvenile or male characteristics is better for identification.

To extract cysts from soil, the commonly used methods are flotation and elutriation. Flotation works on the principle that dried cysts will float. Standard methods include the Fenwick can and Schuiling centrifuge. Elutriation is based on cysts having lower density than soil particles and so can be used for wet soil.

The Fenwick can, as modified by Oostenbrink [19], is the most commonly used instrument for the extraction of cysts from soil samples using the principles of flotation [43,44]. Nematode cysts are relatively light in relation to the inorganic fraction of soil, have a waxy covering, and contain a pocket of air within, so it is possible to separate cysts in the lighter organic fraction of the soil for identification and assessment.

The can tapers toward the top, with a sloping collar around the outside of the rim which collects overflow and directs it towards an outlet. The can has a sloping internal base with a drain plug at its lowest point. Soil is placed at the bottom of the can. Water is then turned on and enters near the bottom of the can. As the can fills, lighter soil particles and cysts flow over the spout and onto sieves from which cysts are “backwashed” after at least 15 min and when the overflow water has become clear.

Soil samples should first be air dried at 37 °C for 48 h to ensure consistency of sample weight and to aid floatation of cysts, which improves efficiency of recovery. If relatively free of organic matter, put the sample of soil directly into the Fenwick can or into a funnel on top of the can. The recommended soil sample size for a smaller or standard-sized Fenwick can (height 30 cm, volume 2 L) is 300 g [45,46]. However, Bellvert [47] found that cyst extraction efficiency was stable in their Fenwick can using soil samples from 100 g up to the physical limits of the can (600 g). Collins et al. [48,49] achieved greater average cyst extraction efficiency using large-scale Fenwick cans (height 50 cm, soil sample size 2 kg) than with medium-sized Fenwick cans (87.5% and 76%, respectively) and concluded that a large Fenwick can is an effective tool for extraction of cysts from large soil samples. Fenwick [43] found very efficient cyst extraction from a can 60 cm high with a capacity of 19 L by using a soil sample size of 4.5 kg.

To achieve improved cyst recovery efficiency, very organic soils should be washed through an 850 μm sieve into the can to allow coarse organic material to be excluded. Fill the can with tap water from the inlet at the bottom, washing through the soil as the can fills. The organic matter with the cysts will rise and overflow onto the collar. Place two sieves with apertures of 850 and 250 μm under the collar outlet. The cysts are collected on the 250 μm sieve for further processing, as they are on average about 450 μm in diameter [46].

## 9. Taxonomic Descriptions

(After Golden and Ellington [50], Stone [29,51], Subbotin et al. [7])


*Globodera pallida*


Female. Body subspherical with projecting neck bearing head, pharynx corpus, isthmus, and anterior part of pharyngeal glands. White in color, some populations passing, after 4–6 weeks, through a cream stage, turning glossy brown when dead. Labial region with amalgamated lips and one or two prominent annuli, deep irregular annulations present on neck, changing to reticulate pattern of ridges over most of body surface. Head framework weakly developed, hexaradiate. Stylet knobs sloping backward. Very large median pharyngeal bulb, almost circular with large crescentic valve plates. Pharyngeal gland lobe broad, frequently displaced anteriad, three gland nuclei. Prominent excretory pore situated at base of neck. Internal structures in neck region often obscured by hyaline secretions on cuticle surface. Vulva a transverse slit at posterior end, set in a slight circular depression or vulval basin. Cuticle surface between anus and vulval basin including about 12 parallel ridges with a few cross connections. Subsurface punctations irregularly arranged over much of body surface, may be confused with surface papillae on vulval crescents.

Cyst. White when first visible on root surface, changing to glossy brown with maturity, subspherical with protruding neck. Vulval region intact or fenestrated with single circumfenestrate opening occupying all or part of vulval basin. Abullate, but small darkened or thickened “vulval bodies” sometimes present in vulval region. Anus visible in most specimens, often at apex of a V-shape mark. Cuticular pattern as in female but more accentuated. Subcrystalline layer absent.

Male. Heat-relaxed specimens C- or S-shaped, posterior part twisted 90–180° about longitudinal axis. Cuticle with regular annulations and four incisures in lateral field, terminating on tail. Labial region offset, rounded with large oral disc, six irregular lips, six or seven annuli, and heavily sc1erotized hexaradiate framework. Stylet well developed with posteriorly sloping basal knobs and cone forming ca 45% of total stylet. Ellipsoid pharyngeal median bulb with strong crescentic valve plates linked by a narrow isthmus encircled by a broad nerve ring, to a narrow, ventrally situated, pharyngeal gland lobe. Hemizonid two annuli long, situated two or three annuli posterior to excretory pore. One testis, commencing with single cap cell 40–65% of body length from head, terminating in a narrow vas deferens with glandular walls. Cloaca with small raised circular lip containing two stout arcuate spicules terminating distally in uni-pointed tips. Small dorsal gubernaculum without ornamentation, slightly wider in dorsoventral aspect. Tail short with bluntly rounded terminus of variable shape.

Juvenile (J2). Lateral field with four incisures but with three anteriorly and posteriorly, occasionally completely areolated. Cuticle thickened for first seven or eight body annuli. Labial region rounded, slightly offset with four to six annuli. Oral disc surrounded by two lateral lips bearing amphidial apertures, adjacent dorsal and ventral submedial lips often fused. Contour of lips and oral disc sub-rectangular [52]. Heavily sclerotized hexaradiate head framework, dorsal and ventral radii bifurcate at tips in 60% of specimens. Stylet well developed, basal knobs with distinct anterior projection as viewed laterally. Gland lobe extending posteriorly for ca 35% of body length. Excretory pore ca 20% of body length from anterior end. Distinct hemizonid two annuli long, located one annulus anterior to excretory pore; hemizonion five or six annuli posterior to excretory pore. Genital primordium at ca 60% of body length from anterior end. Tail tapering uniformly with a finely rounded point, hyaline region forming about half of tail region.


*Globodera rostochiensis*


Female. Pearly white, subspherical to ovate, with elongate, protruding neck. Color changing from white to yellow to light golden as female matures to cyst stage. Cuticle thick, with superficial, rugose, lace-like pattern, D-layer present, punctations resolved near or beneath surface. Labial region slightly offset, bearing two annuli. Labial framework weakly developed. Stylet fairly strong, straight to slightly curved, with well-developed rounded basal knobs, sloping posteriorly. Median bulb large, nearly spherical, with well-developed valve. Pharyngeal glands often obscured but appearing clustered. Excretory pore conspicuous, always at or near base of neck. Vulva terminal, slit of medium length. Vulval area circumfenestrate. No anal fenestration, but anus and vulva both lying in a “vulval basin”, anal area not encircled by cuticular rings. Often beneath vulva, generally in a cluster, are vulval bodies of highly variable size and shape, large superficial tubercles clumped near vulva. Vulva ellipsoid in shape, anus shorter than vulva. All eggs retained in body, no egg mass.

Cyst. Yellow when first visible on root surface, eventually turning brown with age, ovate to spherical in shape with protruding neck, circumfenestrate, abullate, without distinct “vulval bodies” commonly seen in white females. Fenestra circular, anus conspicuous at apex of a V-shaped subsurface cuticular mark. Cyst wall pattern basically as in female but often more prominent, especially near mid-body, tending to form wavy lines going around body. Subcrystalline layer absent. Punctations generally present but variable in intensity and arrangement. Each cyst containing 200–1000 eggs.

Male. Body vermiform, slightly tapering at both anterior and posterior regions. Cuticle with prominent annulation. Labial region slightly offset, hemispherical, with six annuli. Labial framework heavily sclerotized. Stylet strong, with prominent knobs. Anterior and posterior cephalids present. Lateral fields with four equally spaced lines. Median bulb ellipsoidal. Excretory pore ca two annuli posterior to often distinct hemizonid. One testis. Spicules slightly arcuate, tips rounded, not notched. Tail short, variable in length and shape.

Juvenile (J2). Body tapering at both extremities but more at posterior end. Cuticular annulation well defined. Lateral fields with four lines extending for most of body length, outer two lines crenate but without areolation. Labial region slightly offset, bearing 4–6 annuli, considerably wider at base than anteriorly, presenting a rounded, though rather anteriorly flattened, appearance. Labial framework heavily sclerotized. Stylet well developed, with prominent rounded knobs as viewed laterally. Anterior and posterior cephalids present. Valve of median bulb prominent, ellipsoidal. Isthmus and pharyngeal glands typical for the genus. Excretory pore almost adjacent yet slightly posterior to hemizonid. Genital primordium slightly posterior to mid-body, with four cells commonly resolved. Tail tapering to small, rounded terminus. Phasmids generally difficult to see, when visible, located about halfway along tail.

## 10. Identification

*Globodera rostochiensis* and *G. pallida* are morphologically and morphometrically very similar [29,51,52]. Therefore, identification of as many stages as possible should be performed using a combination of morphological characters and molecular techniques.

Nematode cysts separated from soil organic matter must first be carefully inspected using moderate power (up to about 25×) of a dissecting microscope to exclude all non-globose cysts, including those of *Cactodera*, *Betulodera*, *Dolichodera*, *Heterodera*, and *Paradolichodera*.

Any remaining cysts should be considered as suspect PCN cysts. If the laboratory possesses positive control DNA of both species of PCN, single cyst sub-samples should be tested using the PCR protocol provided in Section 10.3.2.

When positive control DNA is not available, there are two potential courses of action, viz. molecular sequencing using the DNA sequencing protocol or morphological examination using the morphological protocol. Morphological identification of suspected *Globodera* cysts and juvenile nematodes to genus and species levels is difficult and requires an experienced nematologist. When a skilled nematologist is available, it is preferable to utilize both the DNA sequencing and morphological protocols to enhance the level of certainty of identification.

### 10.1. Morphological Identification to Genus

An early consideration is how to distinguish cysts of *Globodera* from those of other cyst-forming genera. There is the potential to confuse cysts of *Globodera* with those of the six other genera of the subfamily Heteroderinae, where all females turn into a hard-walled cyst. Cyst shape can be an important character to help distinguish *Globodera* from other genera: globose or spheroid in *Globodera* and generally elongate-ovoid in *Dolichodera* and *Paradolichodera*, and lemon-shaped or pear-shaped in *Betulodera*, *Cactodera*, and *Heterodera*. Occasionally, cysts of *Betulodera* and *Cactodera* tend towards the globose shape, and these specimens can be separated by the presence of a terminal cone, which is a posterior protrusion of the cyst encompassing the anus and vulva and is not present in *Globodera*.

*Punctodera* cysts lack a terminal cone and some species of the genus have globose cysts like *Globodera*, but all can be distinguished from other cyst-forming genera including *Globodera* by the formation of a fenestra in the anal region, of similar shape and size to the vulval fenestra. A fenestra is a terminal region of a cyst where the wall remains very thin and therefore can rupture to permit emergence of juveniles. The vulval slit of *Punctodera* is very short at ˂5 µm, whereas it is about 9 or 10 µm for *G. rostochiensis* [29,50] and about 11.5 µm for *G. pallida* [52]. In cysts of *Globodera*, the anus is at the apex of a conspicuous V-shaped subsurface cuticular mark not seen in *Punctodera*. Additionally, all members of the genus *Punctodera* are parasites of monocotyledonous plants.

Adult female root knot nematodes (*Meloidogyne* sp., family Meloidogynidae), like *Globodera* are swollen and sedentary plant root feeders, and can be distinguished from *Globodera* by their lack of cuticle thickening and pigmentation as a persistent container for the eggs, i.e., a cyst. The perineum of swollen adult female *Meloidogyne* retains its annulation in the form of fingerprint-like whorls, whereas this annulation is lost in *Globodera*. Unlike *Globodera*, female *Meloidogyne* create an egg-mass, which is a collection of extruded eggs embedded within a secreted gelatinous matrix. In addition, females of *Meloidogyne*, but not *Globodera*, are gall-inciting.

Second-stage juvenile specimens of *Globodera* are more robust than their *Meloidogyne* counterparts. The more conspicuous stylet is longer and thicker, and the tail terminus is hyaline (transparent), whereas it is non-hyaline in *Meloidogyne*. The phasmids (paired postanal lateral chemoreceptor sensory organs) of *Meloidogyne* are small and pore-like, whereas they are larger and lens-like in *Globodera*.

Male *Globodera* lack the distinctive lateral amphidial cheeks (outer part of the lateral lip of the head, adjacent the opening of the amphid sense organ) of *Meloidogyne*; they also have a long, slender esophageal isthmus in contrast to the very short, broad isthmus of *Meloidogyne*.

To identify suspected *Globodera* nematodes to genus level, refer to the key in Table 1. Additionally, to identify cysts within the family Heteroderidae, the keys of Hesling [53], Mulvey and Golden [54], Golden [55], Baldwin and Mundo-Ocampo [56], Brzeski [57], Wouts and Baldwin [58], Siddiqi [59], or Subbotin et al. [7] based on cyst form including characteristics of the vulva-anus region, should be consulted.

### 10.2. Morphological Identification to Species

Once all other genera are excluded and it is confirmed that cysts belong to the genus *Globodera*, the following procedure should be followed.

Use a combination of cyst and second-stage juvenile (J2) characteristics if possible. Both stages are normally present in most soil samples infested with PCN, but juveniles will not be extracted by the flotation methods that rely on dried cysts floating to the top of a column of water. Alternatively, to obtain larvae, a cyst can be broken open in a droplet of water on a microscope slide to release the contained eggs. During the process the delicate shells of some eggs will inevitably be broken, enabling the larvae to escape and unfold ready for identification. The most reliable characteristics for identification of second-stage juveniles (J2) within the genus *Globodera* are stylet length, stylet knob width, and stylet knob shape. In *G. pallida*, J2 stylet knobs are distinctly anteriorly directed to flattened anteriorly, and the mean J2 stylet length is >23 µm, whereas in *G. rostochiensis* J2 stylet knobs are rounded to flattened anteriorly, and the mean J2 stylet length is <23 µm (Figure 2).

Cysts should be observed under a dissecting microscope directly on the filter paper used to catch the cysts during the extraction process, at low to moderate magnification (up to about 25×). For species identification, a 40× objective on a compound microscope is adequate to examine the perineal region after the cyst wall has been mounted on a slide. There are no clear differences in size, shape, or color of mature cysts of *G. rostochiensis* and *G. pallida*; the most important cyst differences can be obtained from examination of the perineal area, i.e., number of cuticular ridges between vulval basin and anus (Figure 3), and Granek’s ratio (see Section 10.2.2), the distance from the anus to the nearest edge of the vulval basin divided by vulval basin diameter [60]. However, in some cases cuticular ridges are not visible or are very difficult to count, so Granek’s ratio is considered a more reliable diagnostic tool, and when combined with the important second-stage juvenile measurements, a species diagnosis can be made. Confirmation with molecular techniques is also recommended. 

A key to species of *Globodera* is presented in Table 2. Further keys to species can be found in Mulvey [61,62], Hesling [53], Wouts [63], Golden [55], Wouts and Baldwin [58], Subbotin et al. [7], and EPPO [17].

Three other *Globodera* species could cause confusion during identification of potato cyst nematodes: *G. achilleae* (Golden and Klindic, 1973) Behrens, 1975, *G. artemisiae* (Eroshenko and Kazachenko, 1972) Behrens, 1975, and *G. tabacum sensu lato*. None are parasitic on potato, although the *G. tabacum* species complex (*G. tabacum tabacum* (Lownsbery and Lownsbery, 1954) Skarbilovich, 1959; *G. tabacum solanacearum* (Miller and Gray, 1972) Behrens, 1975, and *G. tabacum virginiae* (Miller and Gray, 1972) Behrens, 1975) parasitizes *Nicotiana tabacum* (tobacco) and some other solanaceous plants (but not potato). To help resolve species determination, Table 3 shows morphometric and morphological comparisons between PCN and *G. achilleae*, *G. artemisiae* and *G. tabacum*. See also Baldwin and Mundo-Ocampo [56], Brzeski [57], Wouts and Baldwin [58], and Subbotin et al. [7,66] for more detailed information on other members of the Heteroderinae.

#### 10.2.1. Microscope Slide-Mounting of Cyst Wall


Place one cyst in a small droplet of water on a glass microscope slide.Puncture the cyst wall with a new scalpel blade towards the neck area to release pressure so further cutting does not cause splits in the cyst wall (Figure 4).Cut across the base of the cyst with a scalpel blade so that a small section containing the perineal region opposite the neck is detached (Figure 5). The smaller the section, the less likely that splits will occur when the wall is flattened in step 7. It is important to avoid creating splits as they can disfigure important diagnostic areas.With a fine needle remove any eggs away from the excised section of cyst wall.Pipette some eggs from the slide surface into an Eppendorf tube for PCR but leave some on the slide for hatching and measuring.Make two more cuts in the cyst section with a scalpel blade as shown in Figure 6.In a small droplet of glycerol on a fresh microscope slide, lay the excised section out flat with the outer surface uppermost (Figure 7). Place a cover slip over the section.Seal the cover slip with clear nail polish, allow to dry, and examine with 40× objective on compound microscope and take measurements in microns.


#### 10.2.2. Taking Measurements for Granek’s Ratio


Measure the distance from the anus at the apex of the V-shaped cuticular mark to the vulval basin (A) (Figure 8).Measure the diameter of the vulval basin (B). Note that there is a “halo” effect around the vulval basin (Figure 8a). The “halo” is not part of the vulval basin. Measurements are taken as shown in Figure 8b.Divide A by B.


### 10.3. Molecular Identification

For potato growers to attain phytosanitary certification, and for a country’s authorities to maintain official control of PCN, molecular techniques are often the preferred choice for regular routine soil testing. When new introductions are suspected, the identification of *G. pallida* and *G. rostochiensis* should combine molecular and morphological methods.

#### 10.3.1. DNA Extraction from PCN Cysts

Cysts collected from soil can be washed/soaked in deionized water or briefly washed in 70% ethanol to avoid possible fungal/bacterial contamination.

The following DNA extraction method works with cysts that contain larvae or unhatched eggs; it does not work with empty cysts.

Material/equipment:DNeasy Blood & Tissue KitPipettes and tips1.5 mL centrifuge tubesTissuLyser (Shaker) machineCentrifuge tube rack/standMicro pestleBalanceCentrifuge with 17,000× *g* capacityShakerRefrigeratorGloves (nitrile)References:

Operation manual of DNA extraction KIT Cat Nos 69504/69506, Protocol for Purification of Total DNA from Animal Tissues (Spin-Column Protocol)

Quader et al. [70]

Method:Place a cyst into 1.5 mL Eppendorf tube and thoroughly crush it with a micro pestle.Add 180 µL ATL (Tissue Lysis) buffer. Gently flick the tube with the pestle to shake off into the buffer as much as possible of the material stuck to the pestle. Close the tube lid and vortex for a few seconds.Add 20 µL Proteinase K. Vortex and briefly spin to get all liquid off the tube lid.Incubate the tube at 56 °C for at least 3 h, vortexing occasionally during this time.Add 200 µL AL buffer. Vortex.Add 200 µL 100% Ethanol. Vortex.Pipette the mixture (including any precipitate) into the DNeasy Mini spin column placed in a 2 mL collection tube (provided). Centrifuge at 6000× *g* for 1 min. Discard flow-through and collection tube.Place the DNeasy Mini spin column in a new 2 mL collection tube (provided), add 500 µL Buffer AW1 (DNA wash buffer), and centrifuge for 1 min at 6000× *g*. Discard flow-through and collection tube.Place the DNeasy Mini spin column in a new 2 mL collection tube (provided), then: (1) add 700 µL Buffer AW2, and centrifuge for 1 min at 6000× *g*. Discard flow-through; (2) add 200 µL Buffer AW2, and centrifuge for 2 min at 16,000× *g*. Discard flow-through and collection tube.Place the DNeasy Mini spin column in a labelled 1.5 mL Eppendorf tube. Use 10 µL tips to pipette any leftover liquid (ca. 2–3 µL) inside the column wall corner immediately above the membrane (this is a poor design of Qiagen columns).Pipette 100 µL Buffer AE (elution buffer) directly onto the DNeasy membrane. Incubate at room temperature for 1 min, and then centrifuge for 1 min at 6000× *g* to elute.Pipette 100 µL Buffer AE directly onto the DNeasy membrane. Incubate at room temperature for 1 min, and then centrifuge for 1 min at 6000× *g* to elute.Measure the extracted DNA concentration where possible.The extracted DNA should be stored at −20 °C until required.

#### 10.3.2. Multiplex PCR for the Identification of Species of PCN

Material/equipment:Pipettes and tipsMyTaq™ Red Mix (Meridian Bioscience)Primers for the nematodes at a concentration of 10 µMDNA marker/ladderSYBR™ Safe DNA Gel StainPCR grade waterThermo-cycler PCR machineAgarose gel and 0.5× TBE bufferIce0.2 mL Eppendorf/PCR tubesTexta/marker penReferences:

Bulman and Marshal [71]

White et al. [72]

PCR primer sequences:

ITS5 5′-CGCGCGGATCCGGAAGTAAAAGTCGTAACAAGG-3′

PIr3 5′-AGCGCAGACATGCCGCAA-3′

PIp4, 5′-ACAACAGCAATCGTCGAG-3′

ITS26 5′-TATATGGATCCATATGCTTAAGTTCAGCGGGT-3′

Primer ITS5 is used in combination with primer PITSr3 in a specific PCR to detect *G. rostochiensis* only. Primer ITS5 is used in combination with primer PITSp4 in a specific PCR to detect *G. pallida* only. Primer ITS5 is used in combination with PITSr3 and PITSp4 to detect both species from a mixed population.

Primers ITS5 and ITS26 should amplify both *G. pallida* and *G. rostochiensis*. These primers are used in a housekeeping nematode PCR to check the quality of DNA extracts. The PCR ensures that DNA is present or that there are no inhibitors in the DNA extracts that retard the activity of the DNA polymerase.

DNA barcoding based on the 18S rDNA gene and the internal transcribed spacer ITS1 region of rDNA (ITS) has been determined as suitable for species identification in *Globodera*. Bulman and Marshall [71] designed the PCR-based *G. pallida*-specific primer PITSp4 and *G. rostochiensis*-specific primer PITSr3 to be used in conjunction with the universal ITS5 primer. These can be used singly or in a multiplex PCR. Alternatively, the universal ITS5 and ITS26 primer pair can be used to amplify the barcoding region, and the resultant product sequenced and compared with verified reference sequences on the NCBS GenBank database.

Method:Determine sample numbers and species of PCN to be tested.Place ice in a suitable container, e.g., esky lid or disposable take away plastic container.Label 0.2 mL PCR tubes according to the number of samples.Make up a master mix of specific PCRs for PCN in a sterile Eppendorf tube by adding the ingredients described in Table 4. Vortex.Make up a master mix of housekeeping PCRs in a sterile Eppendorf tube by adding the ingredients described in Table 5. Vortex.Pipette the master mixes into the PCR reaction tubes.Add PCN DNA templates into each PCR reaction tube. The volume of the DNA extracts can be varied to accommodate 50–100 ng DNA template per reaction.Positive control 1: template DNA of the species *G. rostochiensis*.Positive control 2: template DNA of the species *G. pallida*.Negative control: Sterile distilled water.Spin down all the liquid in the reaction PCR tubes before loading onto the PCR machine.

PCR cycles:Program cycles as showed in Table 6 in PCR machine for 50 µL reaction volumes in case the total reaction volumes are greater than 25 µL.Transfer PCR tubes into PCR machine and start.

Gel run and photograph:Prepare 2% agarose gel (e.g., 2.0 g agarose in 100 mL 0.5× TBE) heated with a microwave oven until agarose is melted and let it cool down to about 70–80 °CAdd a 1/10,000 proportion of SYBR™ Safe (e.g., 1.0 µL SYBR safe in 100 mL melted agarose) and gently swirl agarose solution before pouring into a gel casting tray containing comb(s) and allow to set.Pipette the 7.0–10 µL PCR products into wells.Run PCR products on agarose gel at 100 V for 45 min to 1 h.Visualize and photograph gel under UV light.

PCR product sizes:For Globodera rostochiensis = 434 bpFor Globodera pallida = 256 bpFor housekeeping DNA = 1 kb

#### 10.3.3. DNA Sequencing

For confirmation, the PCR products of the reactions using primers ITS5 and ITS26 for single cysts should be sequenced.

Sequencing reactions using BigDye™ Terminator v3.1 Cycle Sequencing Kit (Table 7) can be done in the laboratory using PCR products cleaned up by QIAquick PCR Purification Kit. Either cleaned PCR products or sequencing reaction products can be sent to Sanger sequencing services, e.g., Macrogen or Micromon, along with the primers to obtain forward and reverse sequences. Forward and reverse sequences of each sample should be de novo assembled and edited/corrected using a suitable computer program, e.g., Geneious. The consensus sequence should be subjected to a database search, e.g., GenBank or private sequence libraries, and phylogenetic analysis. Sequences should be compared with those in GenBank for accession numbers EF622513–EF622532 for *G. rostochiensis* and HQ260426–8, FJ212165 for *G. pallida*. For a match to be positive, the sequence must have a similarity of greater than 99% with these GenBank sequences.

#### 10.3.4. Genotyping

It is possible to compare the genetic differentiation of PCN populations using polymorphic microsatellite DNA markers. DNA can be screened after extracting it from single larvae dissected from cysts. For methodology of this genotyping, see Boucher et al. [73], Alenda et al. [74], and Blacket et al. [75].

There have been many phylogenetic analyses of species within the genus *Globodera* (e.g., [5,73,76,77,78,79,80,81,82,83,84,85,86]). A recent study, based on a phylogenetic analysis of gene sequences of three molecular markers (455 ITS rRNA, 219 *COI*, and 164 *cytb*) of 11 valid and 2 undescribed species of *Globodera* [87], found that *Globodera* displayed two main clades in their phylogenetic trees: (i) *Globodera* from South and North America parasitizing plants from Solanaceae; and (ii) *Globodera* from Africa, Europe, Asia, and New Zealand parasitizing plants from Asteraceae and other families. They hypothesized that the split between solanaceous and non-solanaceous lineages occurred roughly 2.9 ± 0.5 Mya (million years ago), divergence dates of the solanaceous *Globodera* lineages started 2.7 ± 0.2 Mya and the nonsolanaceous *Globodera* lineages 1.6 ± 0.3 Mya, and dispersals of *Globodera* to Europe and New Zealand occurred 1.4 ± 0.3 and 0.9 ± 0.2 Mya, respectively.

## Figures and Tables

**Figure 1 plants-10-00184-f001:**
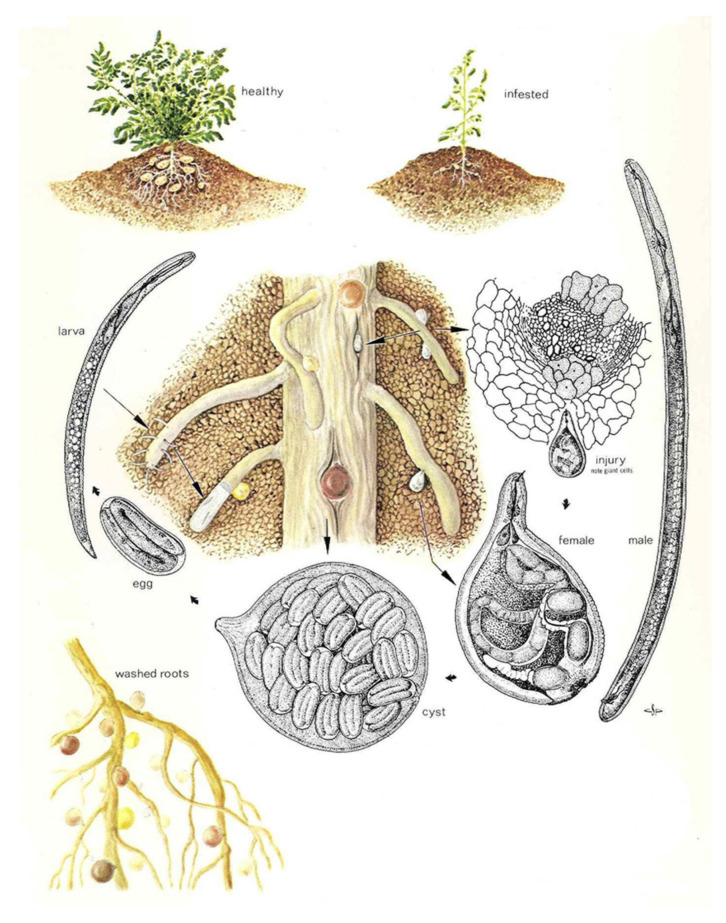
Illustration of the life cycle of *Globodera rostochiensis* (modified after Charles S Papp, Exclusion and Detection, Plant Pest Detection Manual 5:1, California Department of Food and Agriculture, Division of Plant Industry, USA).

**Figure 2 plants-10-00184-f002:**
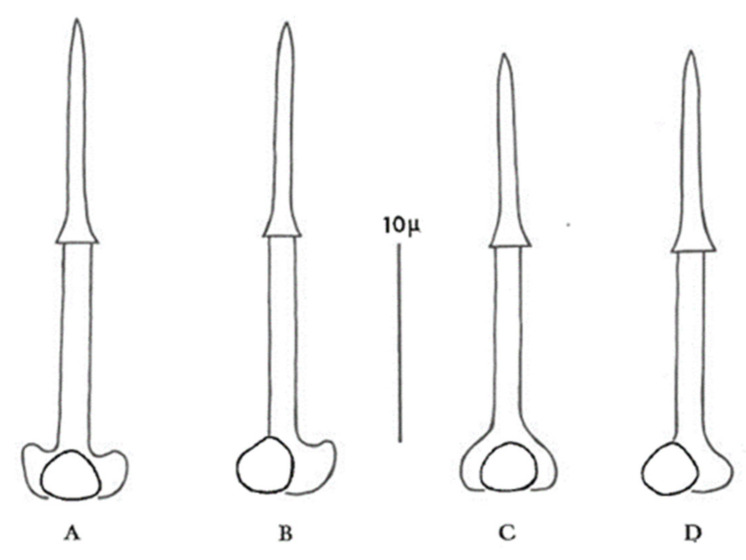
Stylets of second-stage juveniles of *G. pallida* (diagrams (**A**,**B**)) and *G. rostochiensis* (diagrams (**C**,**D**)) (after Stone [52]).

**Figure 3 plants-10-00184-f003:**
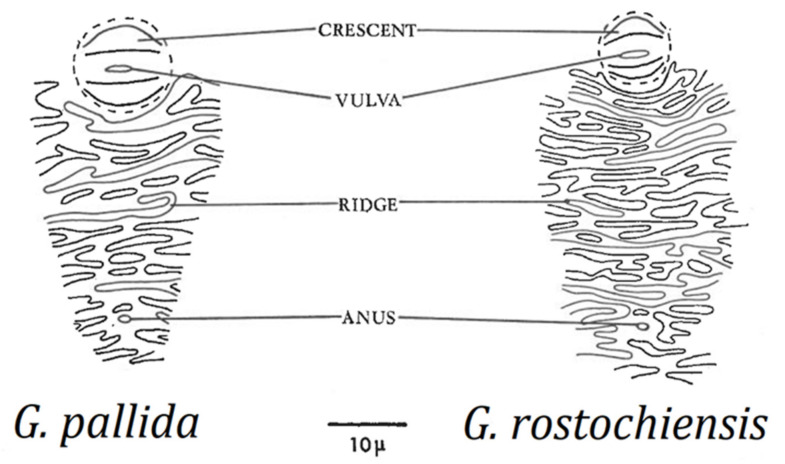
Vulval-anal ridge patterns for *G. pallida* and *G. rostochiensis* (after Stone [52]).

**Figure 4 plants-10-00184-f004:**
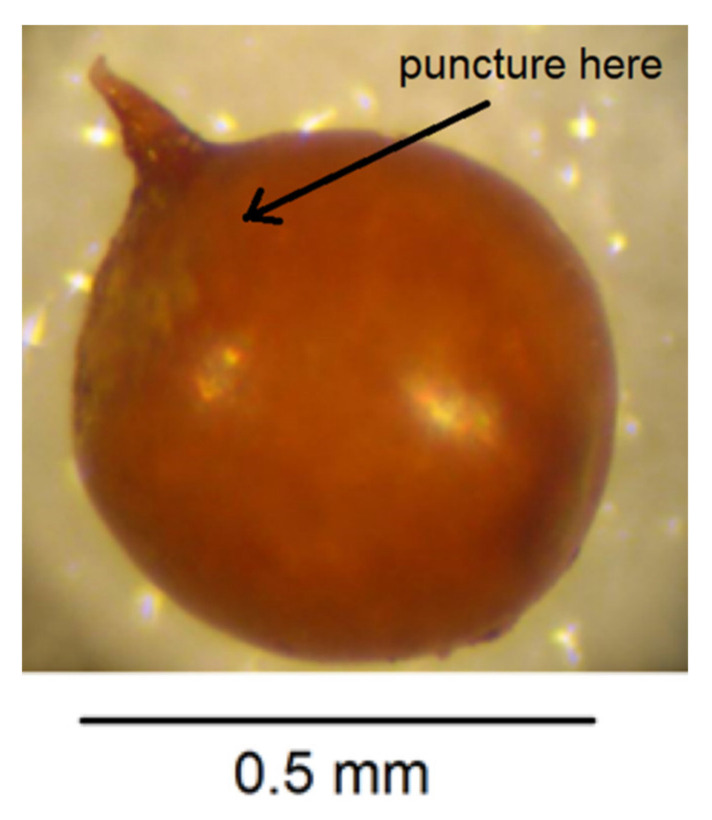
Puncture cyst wall to release pressure.

**Figure 5 plants-10-00184-f005:**
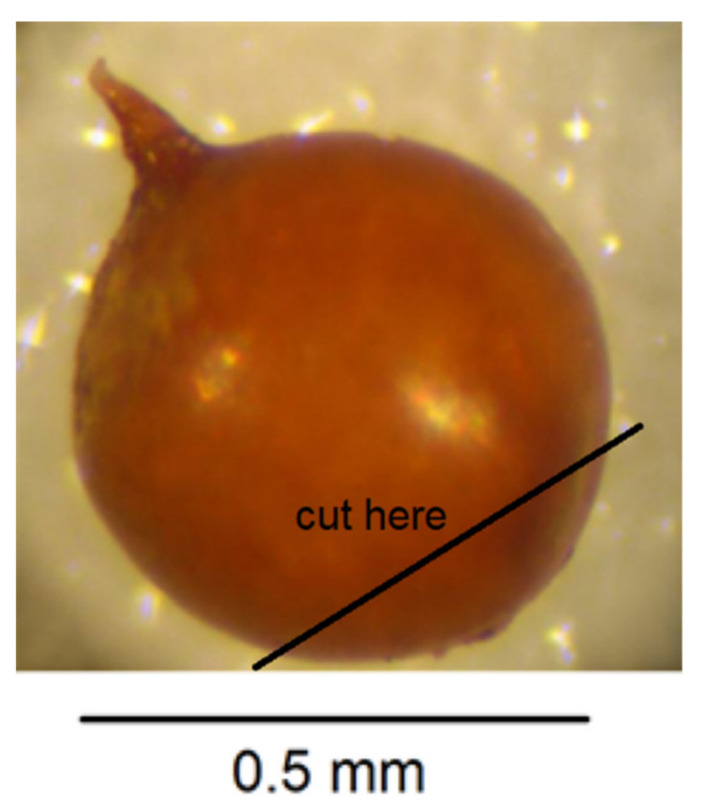
Cut across base of cyst to retain vulval and anal region.

**Figure 6 plants-10-00184-f006:**
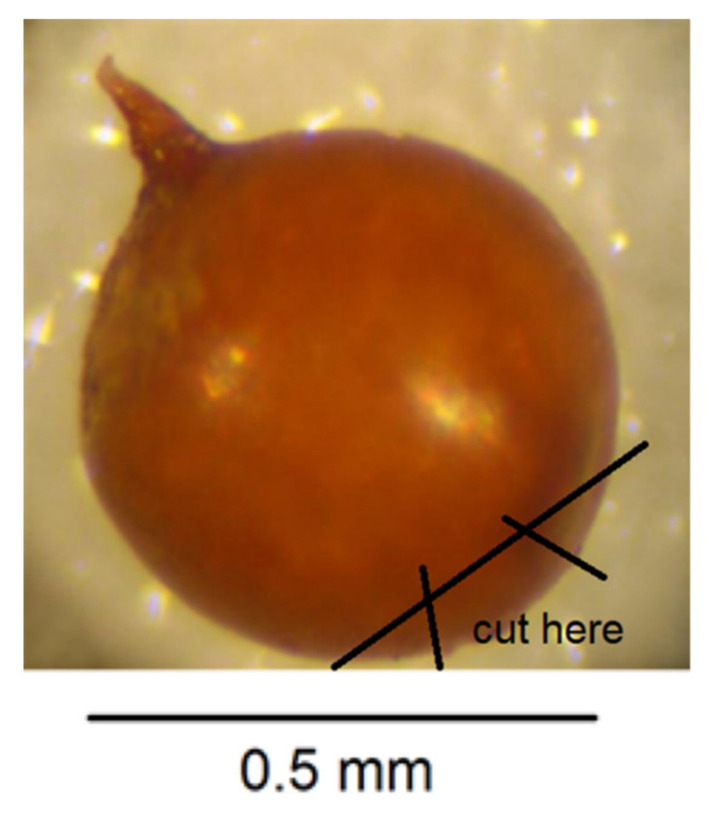
Make two more cuts to the excised section of cyst wall.

**Figure 7 plants-10-00184-f007:**
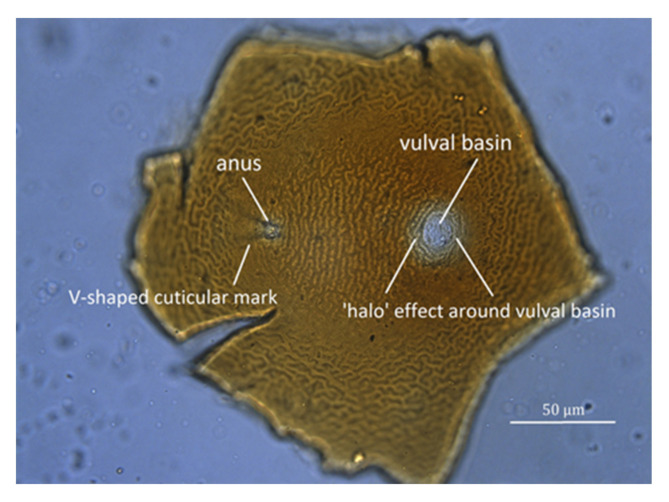
Light microscope image of cuticle surface of perineal region of potato cyst nematode (PCN) (*G. rostochiensis*) cyst laid flat on glass microscope slide.

**Figure 8 plants-10-00184-f008:**
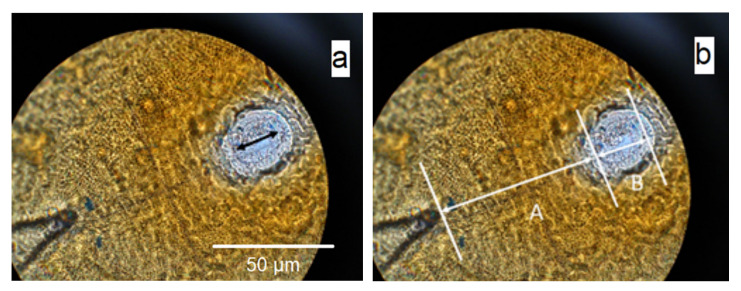
(**a**) Image of perineal region of *Globodera*, showing the “halo” effect around the vulval basin, the actual diameter of which is marked by an arrow. (**b**) Measurements taken to calculate Granek’s ratio, where A is the distance from the anus (at the apex of the V-shaped cuticular mark) to the vulval basin, and B is the diameter of the vulval basin.

**Table 1 plants-10-00184-t001:** Simplified dichotomous morphological key to genus *Globodera*.

1	Nematode with spear or stylet	2
	Nematode without spear or stylet	not *Globodera*
2	Three-part esophagus with a valvulated metacorpus (median bulb) followed by a slender isthmus and glandular basal bulb; stylet with basal knobs	3
	Two-part oesophagus, no valvulated apparatus; stylet usually without basal swelling	not *Globodera*
3	Dorsal oesophageal gland outlet in procorpus; metacorpus less than three-fourths body width	4
	Dorsal oesophageal gland outlet in metacorpus, anterior to valve; metacorpus large, often nearly as wide as body	not *Globodera*
4	Head without setae	5
	Head with setae	not *Globodera*
5	Metacorpus with sclerotized valve	6
	Metacorpus absent or without sclerotized valve	not *Globodera*
6	Mature female greatly enlarged	7
	Mature female vermiform	not *Globodera*
7	Mature female pyriform-saccate, spheroid, or lemon-shaped, usually without tail	8
	Mature female elongate-saccate or kidney-shaped, usually with tail	not *Globodera*
8	Female without irregular body annules around perineum; excretory pore posterior to median bulb; second-stage juvenile stylet usually ˃20 µm; well-developed labial framework	9
	Female with irregular body annules around perineum; excretory pore at level with stylet or close behind it; second-stage juvenile stylet ˂20 µm; weakly-developed labial framework	not *Globodera*
9	Vulva terminal or subterminal; cuticle with lacelike pattern	10
	Vulva subequatorial; cuticle annulated	not *Globodera*
10	Cyst stage present	11
	No cyst stage	not *Globodera*
11	Cyst generally lemon-shaped; vulva on terminal cone	not *Globodera*
	Cyst spherical or subspherical; vulva not on terminal cone	*Globodera*

**Table 2 plants-10-00184-t002:** Dichotomous morphological key to species of the genus *Globodera*. (after Subbotin et al. [7] and EPPO [17], with the addition of *G. agulhasensis* [64] and *G. sandveldensis* [65]).

1	Cuticle of cyst thin, transparent	*G. mali*
	Cuticle of cyst thick, dark in colour	2
2	Mean length of J2 stylet ≤26 µm	3
	Mean length of J2 stylet ≥27 µm	*G. zelandica*
3	Mean length of J2 stylet <19 µm	*G. leptonepia*
	Mean length of J2 stylet ≥19 µm	4
4	Hyaline tail region of J2 >31 µm	5
	Hyaline tail region of J2 ≤31 µm	6
5	Mean J2 body length <500 µm; mean J2 stylet length <24 µm; mean J2 DGO ^1^ <5 µm; mean J2 tail length >58 µm	*G. bravoae*
	Mean J2 body length >550 µm; mean J2 stylet length >26 µm; mean J2 DGO ^1^ >6 µm mean J2 tail length >62 µm	*G. sandveldensis*
6	Mean Granek’s ratio usually >2, mostly parasites of Solanaceae	7
	Mean Granek’s ratio ≤2, mostly parasites of Asteraceae	12
7	With a combination of the following characters: mean J2 DGO ^1^ ≥5.5 µm; mean Granek’s ratio <3; J2 lip region with 4–6 annules; stylet knobs rounded to slightly anteriorly projected	8
	Not with the above combination of all characters; mean J2 DGO ^1^ <5.5 µm	9
8	Cyst wall lacking a network-like pattern, ridges close; mean number of cuticular ridges = 13 (10–18); male spicules with a pointed, thorn-like tip	*G. ellingtonae*
	Cyst wall exhibiting network-like or maze-like patterns; mean number of cuticular ridges = 7–8 (5–15); male spicules with a finely rounded tip	*G. tabacum*
9	Cysts with prominent bullae in the terminal region of most specimens; J2 lip region with 3 annules, mean hyaline tail region >28 µm	*G. capensis*
	Cyst abullate, at most with small vulval bodies in some specimens; J2 lip region with 4–6 annules, mean hyaline tail region <28 µm	10
10	J2 stylet knobs distinctly anteriorly directed to flattened anteriorly; mean J2 stylet length > 23 µm; Granek’s ratio <3	11
	J2 stylet knobs rounded to flattened anteriorly; mean J2 stylet length <23 µm; Granek’s ratio ≥ 3	*G. rostochiensis*
11	Mean Granek’s ratio = 2.1–2.5	*G. pallida*
	Mean Granek’s ratio = 2.8	*G. mexicana*
12	J2 lip region with 5–6 annules	13
	J2 lip region with 3–4 annules	14
13	Mean stylet ≥25 µm in J2, male gubernaculum ^2^ = 11.2–12.9 µm	*G. millefolii*
	Mean stylet ˂25 µm in J2, male gubernaculum = 6.0–9.9 µm	*G. artemisiae*
14	Mean stylet length 26.3 µm; mean DGO ^1^ 5.3 µm; mean hyaline tail region 29.5 µm; number of ridges between anus and vulval basin 6–20	*G. capensis*
	Mean stylet length 23.5 µm; mean DGO ^1^ 4.4 µm; mean hyaline tail region 25 µm; number of ridges between anus and vulval basin 6–12	*G. agulhasensis*

^1^ DGO = distance from anterior end to orifice of dorsal gland opening. ^2^ gubernaculum = grooved cuticular structure which guides the spicule or intromittent organ.

**Table 3 plants-10-00184-t003:** Mean and range (in parentheses) values of some essential characters of *Globodera rostochiensis*, *G. pallida*, *G. tabacum (tabacum)*, *G. achilleae*, and *G. artemisiae,* as given in Baldwin and Mundo-Ocampo [56], Brzeski [57], Fleming and Powers [67], Manduric et al. [68], and Dobosz et al. [69].

	J2 Measurements and Characteristics				Cyst Measurements	
Species	Body Length (µm)	Stylet Length (µm)	Stylet Knob Width (µm)	Stylet Knob Shape	Ridges between Anus and Vulval Basin	Granek’s Ratio
*G. rostochiensis*	468 (425–505)	21.8 (19–23)	(3.2–4.0)	anteriorly flattened to rounded, without forward projections	>14 (16–31)	>3 (1.3–9.5)
*G. pallida*	484 (440–520)	23.8 (22–24)	(4–5)	anterior surface flat to concave with forward projections	<14 (8–20)	<3 (1.2–3.5)
*G. tabacum (tabacum)*	477 (410–527)	23–24	(4–5)	anterior surface rounded	(10–14)	<2.8 (1–4.2)
*G. achilleae*	492 (472–515)	25 (24–26)	(4–5)	anterior surface rounded to anchor shape	<10 (4–11)	1.6 (1.3–1.9)
*G. artemisiae*	413 (357–490)	23 (18–29)	(3–5)	rounded, anteriorly flattened, sometimes slightly indented	(5–16)	1.0 (0.8–1.7)

**Table 4 plants-10-00184-t004:** Master mix of specific PCRs for both PCN species identification in one reaction (multiplex PCR).

Items	Quantity (µL)
Water	10
MyTaq™ Red Mix	12
Primer ITS5 10 µM	1.0
* Primer PITSr3 10 µM	1.0
* Primer PITSp4 10 µM	1.0

* PITSr3 for *G. rostochiensis* and PITSp4 for *G. pallida*.

**Table 5 plants-10-00184-t005:** Master mix of housekeeping PCRs for both PCN species.

Items	Quantity (µL)
Water	10
MyTaq™ Red Mix	12
Primer ITS5 10 µM	1.0
Primer ITS26 10 µM	1.0

**Table 6 plants-10-00184-t006:** PCR cycles for PCN species detection.

Cycles	Temperature °C	Duration
×1 cycle	94	2 min
×35 cycles	94	30 s
	60	30 s
	72	30 s
×1 cycle	72	5 min

**Table 7 plants-10-00184-t007:** Sequencing reaction mix.

Items	Quantity (µL)
Water	13
Cleaned PCR product	60 ng
5× Buffer	3.5
BigDye	1.0
Primer ITS5 10 µM	0.5
Primer ITS26 10 µM	0.5

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
