# Peer review of "Taxonomy, Morphological and Molecular Identification of the Potato Cyst Nematodes, Globodera pallida and G. rostochiensis"

_plants, 2021, doi:10.3390/plants10010184_

Round 1

Reviewer 1 Report

The work by Wainer and Dinh entitled "Taxonomy and Morphological and Molecular Identification of the Potato Cyst Nematodes, Globodera pallida and G. rostochiensis" is well-structured, has a great flow of ideas, and brings important information to the field of nematology, particularly for applied purposes (e.g. PCN diagnosis).

The reviewer has made minor changes/suggestions along the manuscript that the authors should consider prior to resubmitting it.

As a review, the authors should consider to include a small section on the "Phylogeny of Globodera/PCNs" with the inclusion of a phylogenetic tree as well as discussing intra and interspecific genetic variation.  

Finally, the reviewer recommendation is to accept it after minor revisions.

Author Response

Response to Reviewer 1 Comments

Point 1: The reviewer has made minor changes/suggestions along the manuscript that the authors should consider prior to resubmitting it.

Response 1: The following minor changes have been made to the manuscript in accordance with comments of Reviewer 1:

Line 2

Title changed: ‘and’ deleted

Line 12

‘in many parts of the world’ changed to ‘worldwide’

Line 16

‘for’ changed to ‘to’

Line 23

‘they are’ deleted

Line 23

‘be’ inserted

Line 24

‘distinguish’ changed to ‘distinguished’

Line 31

‘in many parts of the world’ changed to ‘crops’

Line 32

Beginning of sentence changed from ‘There are two important species’ to ‘The two important species worldwide are’

Line 34–36

Sentence changed to ‘Globodera species feeding on potato also include G. ellingtonae, restricted to Chile, Argentina and two states in NW USA [3–5] and G. leptonepia (Cobb and Taylor, 1953) Skarbilovich, 1959 found only once in a ship-borne consignment of potatoes [6,7].’

Line 37

‘PCN’ changed to ‘Potato Cyst Nematodes’

Line 37

Comma removed after ‘obligate’

Lines 38–39

End of sentence changed from ‘reduced yields, and sometimes complete crop failure’ to ‘reduce yields, sometimes leading to complete crop failure’

Line 39

‘causes’ changed to ‘cause’

Line 40

‘world, or when’ changed to ‘world when’

Lines 40–42

‘When soil populations of PCN are high, potato yields can be less than tonnage per unit area of the seed planted [9,10] ’ changed to ‘When PCN populations are high in the field, potato yields can be less than the tonnage per unit area of the planted seed [9,10]’

Line 42

‘This severe impact and the’ changed to ‘The severe impact caused by PCN associated with its’

Line 43

‘of PCN’ deleted

Lines 43–44

‘In fact,’ added

Line 50

‘morphologically they are’ deleted

Line 51

‘solely based on morphology’ inserted

Lines 56–59

‘Phylum:                Nematoda, Order: Panagrolaimida, Suborder: Tylenchina, Family: Heteroderidae, Genus: Globodera’ changed to ‘Phylum: Nematoda Diesing, 1861, Order: Rhabditida Chitwood, 1933, Suborder: Tylenchina Chitwood, 1950, Family: Heteroderidae Filip'ev & Schuurmans Stekhoven, 1941, Genus: Globodera (Skarbilovich, 1959) Behrens, 1975’

Line 80

‘Hosts of PCN’ changed to ‘PCN hosts’

Line 86

‘other’ deleted

Line 87

‘act as’ changed to ‘be PCN’

Lines 90–91

‘For more complete lists of hosts of PCN’ changed to ‘For a more complete list of PCN hosts’

Line 92

‘cysts of PCN’ changed to ‘PCN cysts’

Line 93

‘can be’ changed to ‘are’

Line 100

‘move up to about 1 m in the soil and infect by’ changed to ‘can disperse in the soil a distance of about 1 m and infect plants by’

Line 106

‘a’ changed to ‘its’

Line 111

‘Potato cyst nematode’ changed to ‘PCN’

Line 114

‘towards host’ changed to ‘towards the host’

Line 132

‘of potatoes’ deleted

Line 177

‘Fenwick Can’ changed to ‘Fenwick can’

Line 178

‘Centrifuge’ changed to ‘centrifuge’

Line 180

‘Fenwick Can’ changed to ‘Fenwick can’

Line 192

‘it is’ deleted

Line 194

‘Fenwick Can’ changed to ‘Fenwick can’

Line 196

‘Fenwick Can’ changed to ‘Fenwick can’

Lines 197–198

‘Fenwick Cans’ changed to ‘Fenwick can’

Line 198

‘Fenwick Cans’ changed to ‘Fenwick cans’

Line 205

‘Position’ changed to ‘Place’

Line 274

‘ca’ italicised

Line 289

‘DNA’ changed to ‘molecular’

Line 295

‘provided in section 10.3.2’ inserted

Line 299

‘exacting’ changed to ‘difficult’

Line 305

‘member’ deleted

Line 307

‘other’ deleted

Line 344

‘are of’ changed to ‘belong to’

Line 359

‘viewed’ changed to ‘observed’

Line 361

‘25X) or identification to species’ changed to ‘25X). For species identification’

Line 367

‘is a more important’ changed to ‘is considered a more reliable’

Line 369

‘also’ inserted

Line 521 (Reviewer’s suggestion: ‘add some information (one or two sentences) on the molecular marker used for PCN DNA diagnosis: Internal transcribed spacers of the ribosomal RNA (rRNA) gene).

The following paragraph has been added to the manuscript:

DNA barcoding based on the 18S rDNA gene and the Internal Transcribed Spacer ITS1 region of rDNA (ITS) has been determined as suitable for species identification in Globodera. Bulman and Marshall [71] designed the PCR-based G. pallida-specific primer PITSp4 and G. rostochiensis-specific primer PITSr3 to be used in conjunction with the universal ITS5 primer. These can be used singly or in a multiplex PCR. Alternatively, the universal ITS5 and ITS26 primer pair can be used to amplify the barcoding region, and the resultant product sequenced and compared with verified reference sequences on the NCBS GenBank database.

Line 576 (Reviewer’s comment: ‘How important is to sequence these PCR products? What is the general recommendation?)

The following sentence has been modified:

For confirmation, the PCR products of the reactions using primers ITS5 and ITS26 for single cysts should be sequenced.

Point 2: As a review, the authors should consider to include a small section on the "Phylogeny of Globodera/PCNs" with the inclusion of a phylogenetic tree as well as discussing intra and interspecific genetic variation.

Response 2: A brief discussion of the Phylogeny of Globodera/PCNs has been added to section 10.3.4 as follows:

‘There have been many phylogenetic analyses of species within the genus Globodera (e.g. 5, 73, 76–86). A recent study, based on a phylogenetic analysis of gene sequences of three molecular markers (455 ITS rRNA, 219 COI and 164 cytb) of 11 valid and two undescribed species of Globodera [87], found that Globodera displayed two main clades in their phylogenetic trees: i) Globodera from South and North America parasitising plants from Solanaceae; and ii) Globodera from Africa, Europe, Asia and New Zealand parasitising plants from Asteraceae and other families. They hypothesised that the split between solanaceous and non-solanaceous lineages occurred roughly 2.9 ± 0.5 Mya (million years ago), divergence dates of the solanaceous Globodera lineages started 2.7 ± 0.2 Mya and the nonsolanaceous Globodera lineages 1.6 ± 0.3 Mya, and dispersals of Globodera to Europe and New Zealand occurred 1.4 ± 0.3 and 0.9 ± 0.2 Mya, respectively.’

The list of references has been correspondingly updated.

Reviewer 2 Report

This paper describes how to obtain samples of Globodera for identification at the species level. The taxonomic description of the two species of interest are provided together with a morphological key for the genus Globodera. Methods for molecular identification using DNA and PCR are outlined. I have the following comments/suggestions:

Abstract: apart from the final sentence, the abstract is really part of the introduction. I suggest start with the final sentence and then expand to summarise the main points in the paper, providing the reader with an overview of the paper.

line 100: provide a diagram of the life cycle.

lines 203-280: provide diagrams/photos of female, male and juvenile of both species, indicating any differences between them.

line 426, figure 6: I find it difficult to relate figure 6 to figure 7. Can the authors indicate the points shown in figure 7 on figure 6 to show the link between the two. Both figures need a scale. 

Author Response

Response to Reviewer 2 Comments

Point 1: Abstract: apart from the final sentence, the abstract is really part of the introduction. I suggest start with the final sentence and then expand to summarise the main points in the paper, providing the reader with an overview of the paper.

Response 1: The abstract has been extensively modified to contain only a brief summary of the key themes presented in the manuscript.

Point 2: line 100: provide a diagram of the life cycle.

Response 2: A diagram of the life cycle of Globodera has been provided.

Point 3: lines 203-280: provide diagrams/photos of female, male and juvenile of both species, indicating any differences between them.

Response 3:

Adult females: The reviewer would be well aware that mature female cysts of G. rostochiensis and G. pallida do not significantly differ in size, shape or colour – close inspection of the cyst perineal region is the most reliable means to separate the two species. This is clearly shown diagrammatically in figure 2 of the manuscript, viz. vulval-anal ridge patterns for G. pallida and G. rostochiensis.

Juveniles: Similarly, the reviewer would understand that the only unambiguous way to separate larvae of the two species is to examine their styets. Stylets of second-stage juveniles of G. pallida and G. rostochiensis are shown in figure 1. No other morphological features of juveniles allow definitive separation of the two species, so it can be inferred that there is no value in presenting comparative images of juvenile morphological features other than of their stylets.

Males: Furthermore, there are no morphological features of males that permit clear-cut differentiation of G. rostchiensis and G. pallida. It is therefore clear that there is no advantage in presenting comparative images of males of the two species.

Point 4: line 426, figure 6: I find it difficult to relate figure 6 to figure 7. Can the authors indicate the points shown in figure 7 on figure 6 to show the link between the two. Both figures need a scale.

Response 4: Figure 6 has been replaced and labels have been added to show links with figure 7. Additionally, both figures (6 and 7a) have been provided with scale bars.